# Home Language Activities and Expressive Vocabulary of Toddlers from Low-SES Monolingual Families and Bilingual Immigrant Families

**DOI:** 10.3390/ijerph18010296

**Published:** 2021-01-03

**Authors:** Elena Florit, Chiara Barachetti, Marinella Majorano, Manuela Lavelli

**Affiliations:** Department of Human Sciences, University of Verona, Via S. Francesco 22, 37129 Verona, Italy; chiara.barachetti@univr.it (C.B.); marinella.majorano@univr.it (M.M.); manuela.lavelli@univr.it (M.L.)

**Keywords:** low-SES (socioeconomic status), bilingual immigrant families, home language activities, toddlers, expressive vocabulary

## Abstract

Children from low-SES (socioeconomic status) and minority language immigrant families are at risk of vocabulary difficulties due to the less varied and complex language in the home environment. Children are less likely to be involved in home language activities (HLA) in interaction with adults in low-SES than in higher-SES families. However, few studies have investigated the HLA variability among low-SES, minority language bilingual immigrant families. This longitudinal study analyzes the frequency and duration of HLA and their predictive roles for expressive vocabulary acquisition in 70 equivalent low-SES monolingual and bilingual toddlers from minority contexts. HLA and vocabulary were assessed at 24 and 30 months in the majority language (Italian) and in total (majority+minority language) using parent and teacher reports. The frequency and duration of HLA in interaction with adults in total, but not in the majority language, at 24 months were similar for the two groups. These activities uniquely accounted for expressive vocabulary at 30 months, after accounting for total vocabulary at 24 months, in both groups. In conclusion, a minority-majority language context is not an additional risk factor for vocabulary acquisition if HLA is considered in interaction with adults in both languages. HLA are proximal environmental protective factors for vocabulary acquisition.

## 1. Introduction

Bilingual children from low-income, language-minority immigrant families are at risk of language and literacy difficulties [1,2]. These children score below monolingual children from middle-class homes in a range of oral language skills, in particular vocabulary, when tested in one of their languages, especially the majority or societal language (i.e., their second language) [1,3].

The lower levels of vocabulary among bilingual children from low-income, language-minority immigrant families are affected both by socioeconomic status (SES) and by dual language exposure. These are distal environmental factors that have independent and additive effects on oral language [1,4,5]. SES and bilingualism affect children’s language acquisition through a proximal factor that is the quality of the home language environment [6,7,8,9]. Children from low-income families hear less varied and complex language than children from middle-and high-income families [3,6,7] (for reviews see [10,11,12]). Language-minority children’s skill level and development in each language are predicted by the quality of the input, assessed through measures such as the percentage of input provided by native speakers, in dual language environments, even controlling for SES [8,9].

Less is known about how low SES status and dual-language exposure affect the quality of language experience at home measured as the frequency and duration of specific language activities. For instance, shared book-reading and oral story-telling provide children with more communicative interaction and more linguistically diverse and rich input, and are therefore more supportive of language acquisition than activities that provide children with more passive exposure to language, such as television watching [11,13]. On average, children from low-SES language-minority homes are involved to a lower extent in specific language activities than monolinguals from higher-SES homes, but studies of children from low-SES ethnic-minority groups have shown that there is also within-group variability [3,7]. These studies, however, did not involve a group of monolingual majority language children from low-SES homes. As a consequence, it is not clear to what extent low-SES and dual-language exposure in a minority-majority context account for differences and variability. In order to advance research, the first aim of the present study was to compare the frequency and duration of a range of specific home language activities (shared book-reading, oral story-telling, singing, watching TV, watching/playing with smartphone/tablet) in equivalent low-SES monolingual and bilingual toddlers who acquire Italian as the societal or majority language. Evidence from studies of children from low-SES language-minority homes also suggests that specific language activities at home—mainly shared book-reading and oral story-telling in the majority language—predict receptive vocabulary in the same language [3,7]. To extend previous research, the second aim of the present study was to analyze and compare the relation between specific home language activities at 24 months and expressive vocabulary at 30 months in monolingual and bilingual toddlers from low-SES families. For bilinguals, home language activities and vocabulary in the minority and majority language were considered, since both are distributed across the two languages [1,3].

### 1.1. Specific Language Activities at Home

In toddlers and preschoolers, the language input that best promotes language acquisition is characterized by three dimensions. These are interactional features such as responsiveness and shared attention, high levels of linguistic complexity and diversity, as reflected in phonological, lexical, and grammatical features, and the introduction of conceptual content (topics of conversation) such as abstract topics (talk about past or future and unreal events) [10,13,14]. All these features are important supports for language acquisition, with the interactive dimension being especially important for toddlers [13]. Based on this rationale, the specific home language activities considered in the present studies were divided into three types: activities involving interaction with an adult (shared book-reading, oral story-telling, singing); activities with passive exposure to language input (watching TV); and activities involving interaction with digital media (watching/playing with handheld or mobile devices such as smartphone/tablet).

A large number of studies have shown that shared book-reading, oral story-telling, and singing represent ideal conversational contexts in which to provide high-quality input that supports language acquisition, particularly vocabulary acquisition [13,15]. There are several reasons why the frequency and the total time children and parents spend on these activities are related to language acquisition. Firstly, during these activities, language input is directed to the child and contingent on his/her answers or initiatives and may lead to more of those moments of joint attention that promote language acquisition [16,17,18]. Secondly, the language input provided during these activities, especially during shared book-reading and oral story-telling, offers greater lexical diversity and more grammatically rich constructions than conversations produced in other contexts [11,19]. Thirdly, these activities may also lead to more use of decontextualized speech, for example, talking about past or future events and providing causal explanations about how things work in the world [20].

Findings regarding the effects of watching TV on language input and acquisition are mixed [15,21]. Overall, studies either found no relationship or found a negative one between various measures of frequency, duration, and age of first watching TV, and vocabulary acquisition [22,23]. These negative and null results are explained by the lack of social interactions and the fact that the children experience more attentive and socio-cognitive demands than child-directed speech [11,21]. In addition, watching TV implies the lack of opportunity to interact in activities that are more beneficial for language acquisition [22,24].

Studies of the effects on input quality and language acquisition of activities in interaction with digital media are limited since most of the previous studies were carried out before the mass movement toward handheld devices [21]. Toddlers may use these devices to watch video clips, but also to digitally interact using applications available for children from 1–2 to 5 years (e.g., “Prime Parole”/“First words” and “Versi di animali”/“Animal sounds” in the Italian context). These applications provide young children with the possibility to touch icons to hear the names of animals/colors/objects or to touch characters to let them move and hear a nursery rhyme, offering responsive linguistic replies to the infant/child’s actions. In this regard, there is evidence that, when videos are contingent and responsive to the infants’ behaviors, infants can learn a language [24]. Roseberry and colleagues [24] showed that toddlers aged 24–30 months learn novel words only in contingent interaction with an adult or with a video chat but not in noncontingent video watching. These results suggest that interactions with handheld or mobile devices may be more similar to activities involving interaction with an adult than activities with passive exposure to a language. As a consequence, the content of handheld or mobile devices may promote language acquisition. It should be noted, however, that studies on the effects of interactive elements in electronic stories showed that these elements are beneficial provided that they are not distracting and do not disrupt language comprehension and learning [25].

### 1.2. Language Activities at Home in Low-SES and Minority-Majority Language Contexts 

Parents from low-SES families produce language input characterized by less varied vocabulary, less complex syntax, and more directive speech than higher SES parents in a variety of contexts (i.e., toy play, mealtime, shared-book reading) [6,11,26,27]. Parents with less income and education are less likely to read to their children than middle-high income parents and to engage their preschool-age children in fewer challenging discussions (e.g., involving inference during reading) [28,29,30,31,32]. Finally, there is evidence that variability in SES is related to the occurrence of additional home language activities such as oral story-telling and singing [3]. In sum, there is ample evidence that children from low-SES homes received less varied and complex input, especially through home activities most relevant to support language acquisition. 

Recent evidence suggests that there is also much variability in the frequency of home language activities—shared book-reading and oral story-telling—among U.S. children from low-SES backgrounds [7,14,33]. For instance, many U.S. toddlers and preschoolers from low-income ethnic-minority families were read to relatively often by their parents (daily or a few times a week), and involved in oral story-telling interactions and high-quality reading [7,33]. 

Studies of children from U.S. low-income ethnic-minority families have provided information on factors that may account for additional effects of dual-language exposure on home language activities in low-SES backgrounds [7,33,34]. These factors relate to the material and human resources of the family. Language use at home in low-SES ethnic-minority families who primarily spoke English or Spanish, or spoke both languages, was related to the availability and variety of narrative books at home. In particular, children of English-speaking parents had about 20 more books and a greater variety of narrative books than did children of Spanish-speaking parents [34]. This result showed that non-English-speaking families have less access than English-speaking families to narrative books that require knowledge of the majority language and culture. This, coupled with the lack of books in the families’ home language [35], makes it less likely that the children will be involved in home language activities. The variety and number of books in the home, for instance, were positively related to the range of reading experiences and to the frequency of shared book-reading and oral story-telling with mothers [33,34]. 

In sum, children from low-SES homes are less likely to be involved in language activities such as shared book-reading and oral story-telling than children from middle-high SES homes. Nevertheless, there is variability in the home language activities carried out by parents from low-SES families and a minority-majority language context (where the minority language is likely to be the primary language at home) may account for additional effects on specific home language activities. However, previous studies did not include a group of monolingual children from low-SES families for comparison with bilinguals with equivalent SES, nor have they assessed home language activities other than those involving interaction with an adult. Finally, most of the literature so far has assessed home language activities in general without considering which language(s) (the minority and/or the majority language and/or total language) was/were used in home language activities [33,34]. This is a significant gap because it is important to have a fuller picture of home language activities in low-SES bilingual children, as the linguistic input is divided over the two languages [3]. 

### 1.3. Relations between Home Language Activities and Children’s Vocabulary in Low-SES and Language-minority Families

Malin and colleagues [7] analyzed the longitudinal relationship between the frequency and the quality (i.e., metalingual talk during reading) of shared-book reading of mothers and fathers from low-income ethnic-minority families with their two-year-old children and children’s receptive vocabulary in English (majority language) in pre-kindergarten. The quality of shared book-reading, but not the frequency, was related to later receptive vocabulary in pre-kindergarten, controlling for parental education. The frequency of singing and story-telling (composite score) in the minority and majority languages correlated with the receptive vocabulary scores of low-SES Moroccan-Dutch and Turkish-Dutch 3-year-olds in both languages, controlling for SES [3]. The pattern of relations of a monolingual Dutch control group showed that the frequency of story-telling and reading (composite score) correlated with receptive vocabulary. SES differences between the monolingual and bilingual groups, however, were not controlled for. Taken together, these findings show that activities involving interaction with an adult are related to the receptive vocabulary of toddlers and preschoolers from low-SES and language-minority homes. 

Scheele et al. [3] did not find that the frequency of TV-watching in the minority or majority language was related to receptive vocabulary in either the monolingual and bilingual groups. This result is in line with the wider literature (see Section 1.1.) suggesting that passive exposure to language through television watching does not support language acquisition.

Finally, a meta-analysis by Takacs et al. [25] provided some evidence that preschool children’s independent interactions with technology-enhanced stories presented through a variety of digital devices (e.g., computer, tablet) promoted the learning of expressive vocabulary to a greater degree than interactions with more traditional storybooks, for groups of children at risk of language difficulties for different factors (e.g., low-SES, immigration, language delay) but not for children not at risk of language difficulties. This result suggests that interactions with digital devices may have positive effects on vocabulary. It should be noted, however, that children at risk of language difficulties profited more from multimedia features embedded in technology-enhanced stories (e.g., animated pictures, sounds, and music) if they were congruent with the narration, but were also more distracted by interactive features than children not at risk of language difficulties. The authors concluded that interactive features in technology-enhanced stories were distracting, especially when there were many possibilities for interaction.

In sum, previous studies of low-SES and language-minority homes analyzed the relations between home language activities and receptive vocabulary. A monolingual low-SES group and assessment of language activities and vocabulary skills in both minority and majority languages were not consistently included. 

### 1.4. The Present Study

The present study addressed two specific aims:
In toddlers from low-income, monolingual and language-minority immigrant families to compare the frequency and duration of home language activities (a) involving interaction with an adult (shared book-reading, story-telling, and singing), (b) with passive exposure to language input (watching TV), and (c) involving interaction with digital media. We expected that monolingual parents involve children in home language activities in the majority language more frequently than do bilingual parents. We expected this especially for activities in interaction with an adult. When home language activities in total (i.e., in the minority + majority language) were considered, the overall differences were expected to be reduced [3];To analyze the longitudinal relations between each of the three types of home language activities—involving interaction with an adult, with passive exposure to language input, and involving interaction with digital media—at 24 months (Time 1) and expressive vocabulary at 30 months (Time 2). The home language activities at Time 1 and expressive vocabulary at Time 2 were assessed both in total language (minority + majority language) and in the majority language. In analyzing the longitudinal relation between home language activities at Time 1 and expressive vocabulary at Time 2, group and total vocabulary at 24 months were controlled for. We expected that:-the frequency and duration of home language activities involving interaction with an adult would be related to later expressive vocabulary when considering home language activities in total [3,7];-neither the frequency and duration of activities with passive exposure to language input nor the frequency and duration of activities involving interaction with digital media would account for expressive vocabulary, whether in total language or in the majority language [3];-group (i.e., toddlers belonging to monolingual vs. language-minority immigrant families) would account for expressive vocabulary in the majority language but not for expressive vocabulary in total (i.e., minority + majority language).


## 2. Materials and Methods

### 2.1. Participants

Seventy 24-month-old children (Time 1) regularly attending five nursery schools in a northeastern province of Italy participated in the study. The children were divided into two groups: 46 children from bilingual immigrant families (57% females) and 24 children from Italian families (46% females). The children from bilingual immigrant families were mainly exposed to Italian and Romanian (26 children; 57%) or to Italian and Nigerian English (9 children; 20%); five children were exposed to Italian and Sinhalese, and; the remaining children to Italian and Arabic (3), Spanish (2), and Portuguese (1). The distribution of the immigrant families is consistent with the population demographic of the province. The monolingual children were exposed only to Italian.

The participants involved in the present study were selected from those recruited for a larger longitudinal study on the lexical trajectories of children from low-income families. Only low-income families—identified as those paying the lowest rate for their children’s nursery schools (≤€130 per month)—were selected for participation in this study. All the children were born in Italy within two weeks of their due date and they were healthy at birth, with normal hearing. Children with certified disabilities or developmental disorders were not included in the sample.

The parents (usually mothers) of all the children, monolingual and bilingual, were involved in the collection of information on the family’s demographics and language exposure (see Section 2.2 for more information). Based on the parents’ reports, for the great majority of the bilingual children (85%), the less-frequently heard language constituted at least 30% of their language exposure (*M* = 49.77%, *SD* = 13.33); for the remaining children, the less-frequently heard language was between 23–28% of their language exposure (*M* = 24.86%; *SD* = 2.08). All the children were exposed to each language every day from birth or sometime later (*M* = from 5 months after birth). Based on these data, we considered bilinguals in the present study as ‘bilingual first language acquisition’ or ‘early bilinguals’ [36,37].

All the parents of the bilingual group were native speakers of the main language of their country: Romanian (26), Nigerian English (9), Sinhalese (5), Arabic (3), Spanish (2), and Portuguese (1). Two Nigerian parents stated that they also spoke the language of their ethnic group (Edo) but never directly to their children. All the parents of the monolingual group were native Italian-speakers.

The demographic characteristics of the children and their mothers are reported in Table 1.

No significant differences (the *p*-value was adjusted to 0.007 after Bonferroni correction for multiple comparisons) were found between the bilingual and monolingual groups in the children’s gender, birth order, singleton condition, age of entry, and daily attendance at nursery school, maternal formal education, and age.

All the participating nursery schools were state-regulated and funded. All the children were cared for in a group of up to 8 children. A total of 28 nursery teachers primarily responsible for the participating children participated in the study and evaluated the children’s vocabulary in the majority language (i.e., Italian; see Section 2.2); sixteen of them evaluated more than one child. All the teachers were female native Italian-speakers and participated in 4 h of ad hoc training before starting the evaluation of children’s vocabulary. Six teachers held a degree in Education and 22 of them held a high school diploma in Infant Education or Teaching. The study was approved by the host university’s ethics committee (ethical approval code: VOCALIF, Cod. 2018_5). 

### 2.2. Procedure

The present study adopted a two-waves longitudinal design spanning a six-month period. A multimethod approach combing the use of questionnaires, semi-structured interviews, and standardized instruments was used.

At the beginning of the study, parents were asked to fill out a consent form and a demographic questionnaire aimed at collecting various information concerning the families (e.g., children’s and parents’ age, parents’ marital status, and education). Around the child’s second birthday (Time 1), trained researchers collected data on the family’s home language activities and the children’s vocabulary skills in both the majority language and, in the case of the bilinguals, the minority language. Both parents and nursery teachers were involved as informants. The parents of the monolingual participants completed a semi-structured interview on home language activities at home. A semi-structured interview with the parents of immigrant families was carried out in a quiet room at the nursery school in order to collect data on home language activities in the minority and majority language. The interviews were conducted in Italian and, when necessary, with the help of a cultural mediator in the language of the parent’s choice. At the end of the interview, bilingual parents were asked to complete the CDI-Words and Sentences-short form: the Italian version, translated into their language (see details in Section 2.3.2). The nursery teachers were required to complete the Italian version of the CDI, on their own. At 30 months (Time 2), following the same procedure, the bilingual parents and the teachers were required to complete the CDI short form adapted to the language-minority and the Italian version of the CDI, respectively.

### 2.3. Measures

#### 2.3.1. Semi-structured Interviews on Home Language Activities (Time 1)

The semi-structured interviews on home language activities were constructed based on the instruments devised by Onofrio et al. [38] to analyze the linguistic profile of bilingual children and by Vander Woude and Barton [39] to analyze shared book reading activities.

The parents were asked for information about the frequency and duration of home language activities during the month preceding the interview. Questions asked for the frequency of (a) activities implying interaction with an adult (i.e., shared book-reading, oral story-telling, singing); (b) activities implying passive exposure to language input (i.e., watching TV); and (c) activities implying interaction with digital media (i.e., playing with mobile devices such as smartphone and tablet). Four-point Likert scales (0 = never, 1 = twice a month, 2 = twice a week, 3 = every day) were used as answer formats. Parents also provided information about the approximate typical duration of shared book-reading, watching TV, and playing with smartphone/tablet. Information about the duration of oral story-telling and singing were not collected because of parents’ difficulties at quantifying the time spent in these activities. Parents from language-minority immigrant families were asked for the frequency of the three types of activities in the majority and minority languages. They were also asked for the approximate typical duration of the different types of activities carried out in the majority and minority languages. Nevertheless, the duration of shared book-reading was not assessed separately in each language because of parents’ difficulties at differentiating this information between the two languages.

The analyses, frequencies, and time devoted to each activity were numerically transformed on a weekly basis. As a consequence, frequencies were expressed as: 0 (never), 0.5 (twice a month), 2 (twice a week), 7 (every day). The time devoted to home language activities was transformed by multiplying each value by frequencies on a weekly basis and expressed in hours. Measures of home language activities in the majority language and in total language (minority+ majority language) were obtained. For frequencies of activities implying interaction with an adult (i.e., shared book-reading, oral story-telling, singing), an average score was computed.

#### 2.3.2. Expressive Vocabulary (Time 1 and Time 2)

Expressive vocabulary in the majority language was assessed using the Italian version of the CDI: Words and Sentences-short form (Primo Vocabolario del Bambino-PVB) [40,41]. The short form of the CDI includes a 100-word expressive vocabulary checklist with lexical categories such as nouns (animals, toys, food, people), verbs, adjectives, adverbs, and closed-class words. Teachers were asked to report which words children were able to produce from the 100-item list, excluding imitations or elicited repetitions. The vocabulary skills in the minority language were assessed using the Italian CDI-short form translated into Romanian, Nigerian English, Sinhalese, Arabic, Spanish, and Portuguese by cultural mediators in collaboration with a language development researcher. All versions were 100 words long and were developed for the participants of the present study. Because the translation of each instrument into another language is problematic [42], the cultural mediators involved in the translation process were language-minority native speakers so that they could adapt the words of the checklist according to the different cultures. The total number of words checked by parents and nursery teachers yielded the children’s expressive vocabulary scores in the minority and majority language, respectively. In the analyses, expressive vocabulary in the majority language and in total language (i.e., minority + majority language) were considered. 

### 2.4. Data Analysis

First, preliminary analyses were carried out for identifying outliers and exploring values of skewness and kurtosis. Second, four mixed ANOVAs with types of language activities as the within-subject factor (3 levels: activities involving interaction with an adult, passive exposure to language input, and interaction with digital media) and groups as the between-subject factor (2 levels: monolingual and bilingual) were carried out in order to compare frequency and duration of home language activities in monolingual and bilingual immigrant families (aim 1), considering both the total language (i.e., minority + majority language) and the majority language. Significant interactions were explored with *t*-tests with Bonferroni’s correction for multiple comparisons. 

Finally, correlations (controlling for groups) and regressions were carried out in order to analyze relations between the three types of home language activities and expressive vocabulary at 24 and at 30 months (aim 2). Given that maternal formal education is a relevant factor in the power of SES to predict children’s language skills [10], this variable was also considered as a control in correlation analyses. Correlations were carried out by considering home language activities in total language (i.e., minority+ majority language) at 24 months, and expressive vocabulary both in total and in the majority language at 30 months. Linear hierarchical regressions were used in order to identify, among home language activities, the longitudinal predictors of children’s expressive vocabulary at 30 months, after controlling for the effect of group and total expressive vocabulary at 24 months. 

## 3. Results

### 3.1. Home Language Activities of Toddlers from Low-SES Monolingual and Bilingual Immigrant Families (Aim 1)

Preliminary analyses did not detect extreme outliers, and the values of skewness and kurtosis for all the variables related to home language activities were all within acceptable limits [43].

Table 2 shows the descriptive statistics for frequency and duration of home language activities on a weekly basis in total (i.e., majority + minority language) and in the majority language for bilingual and monolingual families. 

On average, children in both groups were involved in total language activities involving interaction with an adult more than twice a week. The details were as follows: 17% and 25% of bilinguals and monolinguals, respectively, were involved every day; 72% and 71%, respectively, were involved about twice a week; 11% of bilinguals and 4% of monolinguals were involved no more than twice a month. However, 13% of bilingual children were involved no more than twice a month in activities involving interaction with an adult in the majority language, and 9% of them were never involved. In both groups, there was a wide range of frequency and some contrasting behaviors in home language activities with passive exposure to language input (i.e., watching TV) and involving interaction with digital media (i.e., playing alone with smartphone/tablet). Indeed, the majority of bilingual (89%) and monolingual (67%) children watched TV in the majority language almost every day, while the remaining 11% of bilinguals and 33% of monolinguals never watched TV. Two-thirds of the bilingual children and one-third of the monolingual children watched or played alone with smartphone/tablet almost every day, while one-third of bilinguals and two-thirds of monolinguals were never involved in activities interacting with digital media.

The mixed ANOVA on the frequency of home language activities in total showed a main effect of type of activities [*F*(1.75,118.94) = 8.78, *p* = 0.001, *η_p_*^2^ = 0.114] and group [*F*(1,68) = 7.40, *p* = 0.008, *η_p_*^2^ = 0.098)] which was qualified by a significant interaction between type of activities and group [*F*(1.75,118.94) = 5.63, *p* = 0.004, *η_p_*^2^ = 0.076)]. Independent *t*-tests with Bonferroni correction for multiple comparisons (*p* < 0.016) showed that bilinguals engaged in practices involving interaction with digital media more frequently than monolinguals [MBIL−MMONo = 2.39, *t*(68) = 2.84, *p* = 0.006] but the two groups did not differ for the frequency of practices interacting with an adult [MBIL – MMONo = −0.60, *t*(68) = −1.34, *p* = 0.184] or with passive exposure to language input [MBIL – MMONo = 1.57, *t*(33.54) = 2.07, *p* = 0.05]. The mixed ANOVA on the frequency of home language activities in the majority language showed a main effect of type of activities [*F*(1.75,119.26) = 15.45, *p* = 0.001, *η_p_*^2^ = 0.185] which was qualified by a significant interaction between type of activities and group [*F*(1.75,119.26) = 6.69, *p* = 0.002, *η_p_*^2^ = 0.090]. The main effect of group was not significant [*F*(1,68) = 0.54, *p* = 0.465, *η_p_*^2^ = 0.008]. Independent *t*-tests (*p* < 0.016) showed that monolinguals engaged more frequently in practices involving interaction with an adult in the majority language compared to bilinguals [MBIL − MMONo = −1.66, *t*(68) = −3.64, *p* = 0.001] but the two groups did not differ for the frequency of practices with passive exposure to language input [MBIL − MMONo = 1.57, *t*(33.54) = 2.07, *p* = 0.05] and interaction with digital media [MBIL − MMONo = 1.02, *t*(48.77) = 1.18, *p* = 0.246]. 

The mixed ANOVA on the duration of home language activities in total showed a main effect of type of activities [*F*(1.71,114.35) = 27.28, *p* = 0.001, *η_p_*^2^ = 0.289] and group [*F*(1,67) = 9.38, *p* = 0.003, *η_p_*^2^ = 0.123], which was qualified by a significant interaction between type of activities and group [*F*(1.71,114.35) = 4.08, *p* = 0.025, *η_p_*^2^ = 0.057]. Independent *t*-tests (*p* < 0.016) showed that bilinguals engaged for longer than monolinguals in practices that implied passive exposure to language input [MBIL – MMONo = 3.8, *t*(61.52) = 3.72, *p* = 0.001] but the two groups differed neither in the time devoted to practices that implied interaction with adult [MBIL – MMONo = −0.10, *t*(67) = −0.30, *p* = 0.768], nor in the time devoted to interaction with digital media [MBIL – MMONo = 2.72, *t*(68) = 1.64, *p* = 0.105]. The mixed ANOVA on the duration of home language activities in the majority language showed a main effect of type of activities [*F*(1,68) = 10.86, *p* = 0.002, *η_p_*^2^ = 0.138] and group [*F*(1,68) = 5.20, *p* = 0.026, *η_p_*^2^ = 0.071]. Interaction between type of activities and group was not significant [*F*(1,68) = 0.946, *p* = 0.334, *η_p_*^2^ = 0.014]. A paired *t*-test showed that children engaged in activities with passive exposure to language input for longer than in practices interacting with digital media [MPAS = 7.60 and MDG = 3.98; *t*(69) = 3.80, *p* = 0.001]. An independent *t*-test showed that bilinguals engaged for longer than monolinguals in activities with passive exposure to language input and interaction with digital media [MBIL = 9.74 and MMONo = 6.46; *t*(68) = 2.87, *p* = 0.005].

### 3.2. Longitudinal Relations between Home Language Activities at 24 Months and Vocabulary at 30 Months (Aim 2)

Table 3 shows the descriptive statistics for expressive vocabulary size (number of words) at 24 and 30 months. Values of skewness and kurtosis for vocabulary variables were all within acceptable limits [43].

Table 4 reports partial correlations (i.e., Pearson’s *r* controlling for the group) between home language activities in total and in the majority language, and maternal formal education at 24 months, and expressive vocabulary—considering both total vocabulary and majority language vocabulary—at 24 and 30 months. 

The results showed that the intra-correlations between frequency and duration of home language activities in total and in the majority language were large. The frequency of activities involving interaction with an adult in total was significantly correlated with expressive vocabulary in the majority language at 30 months. The duration of shared book-reading in total was significantly correlated with total vocabulary and expressive vocabulary in the majority language at 30 months. Maternal formal education was correlated neither with expressive vocabulary in total language at 30 months, nor with expressive vocabulary in the majority language at 30 months, and was not, therefore, controlled for in the next analyses.

Fixed-order hierarchical multiple regression analyses were carried out in order to identify the longitudinal predictors of total vocabulary and expressive vocabulary in the majority language at 30 months. Table 5 and Table 6 show the results of the regression analyses. 

The criterion variable of the first and second regression analyses was total vocabulary at 30 months (Table 5). In the first and second regressions, group and total expressive vocabulary at 24 months were included as predictors in the first step (Model 1). The frequency of activities involving interaction with an adult and duration of shared book-reading in total at 24 months were included as predictors at step 2 (Model 2) in the first and in the second regressions, respectively. The amount of variance explained by the first and second regressions was 36%. The first and second regressions showed that frequency of activities in interaction with an adult, and duration of shared book-reading in total at 24 months, uniquely accounted for total expressive vocabulary at 30 months when group and total expressive vocabulary at 24 months were taken into account. Total expressive vocabulary at 24 months, but not group, was a significant predictor.

The criterion variable of the third and fourth regression analyses was expressive vocabulary in the majority language at 30 months (Table 6). In both regressions, group and expressive vocabulary in total at 24 months were included as predictors in the first step (Model 1). The frequency of activities involving interaction with an adult and the duration of shared book-reading in total at 24 months were included as predictors at step 2 (Model 2) in the third and fourth regressions, respectively. The amount of variance explained by the third and fourth regressions ranged from 50% to 52%. Both the frequency of activities implying an interaction with an adult and the duration of shared book-reading in total at 24 months uniquely accounted for expressive vocabulary in the majority language at 30 months, over and above group and expressive vocabulary in total at 24 months. Total expressive vocabulary at 24 months and group were significant predictors. 

## 4. Discussion

The present study sheds light on the effects of a minority-majority language context on the quality of toddlers’ language activities at home, and the contribution of these activities to expressive vocabulary acquisition. It extends the existing literature by addressing two main aims: first, comparing toddlers from language-minority immigrant families and monolingual families matched for low-SES on the frequency and duration of a range of home language activities, including activities involving interaction with adults, watching TV, and consumption of digital media; second, investigating the predictive role of these activities on children’s expressive vocabulary assessed six months later. Unlike most of the previous research, home language activities and expressive vocabulary in both the majority or societal language and in total (minority + majority language) were analyzed in toddlers living in Italy, a sociocultural context relatively unexplored.

### 4.1. When Considered in Total, Home Language Activities with Adults Are Similar (and few) in Low-SES Monolingual and Bilingual Immigrant Families

The results supported our expectations. Indeed, the frequency and duration of total home language activities involving interaction with an adult were similar for both minority-language immigrant and monolingual children but this was not true in the case of majority language only. This result may be explained by considering that immigrant parents are likely to interact with their children in the minority language because the minority language reflects their cultural identity. In this sense, they feel that it is their duty to transmit the heritage language to their children in the face of an environment where the dominant language reflects another cultural model. However, migration and acculturation imply a complex, multidimensional process of change and continuity of parenting practices, values, and identification of the heritage culture as well as of the receiving culture [44,45], a process that also shapes the home language environment [33,34]. Therefore, immigrant parents belonging to different ethnic minorities may use the minority language with their children to a different extent. In our sample, for instance, Sinhalese and Moroccan mothers emphasized the importance of using the minority/heritage language with their children; in contrast, several Romanian mothers reported their efforts to interact with children in the majority language. 

Another factor that may contribute to explain the results is that immigrant parents are native speakers in a minority language, and they vary in their majority language proficiency. They might, therefore, feel uncomfortable speaking to their children in a language they do not know well, and so be less willing to share language activities such as book-reading, given that the books available for children in the host country are in the majority language [33]. In this regard, it is interesting to observe that oral story-telling—a home language activity that requires nothing more than the caregiver’s personal skills—was reported in bilingual immigrant families to be performed more frequently in the minority language than in the majority language. Furthermore, it is interesting to note that none of the immigrant parents reported that they never involve children in language activities interacting with adults, but 9% of them reported that they never involved children in majority-language activities. 

Descriptive statistics, however, showed that there was a wide range in both groups in the frequency of home language activities involving interaction with an adult. These findings are generally consistent with past studies that highlight variability among low-SES families in the U.S. [7,33] and they extend the literature both by looking at younger children living in Italy and by comparing language activities in total with activities in the majority language. Notably, although there was also some variation in the duration of shared book-reading, about half of the parents (55% and 54% for the bilingual and monolingual group, respectively) reported that they read to their children for at most one hour a week. This result, that children from low-SES families are not involved in this activity for long, is consistent with results reported for low-SES families living in the U.S. [27].

Unlike activities involving interaction with an adult, activities with passive exposure to language input (watching TV) and in interaction with digital media were absent in a relatively small percentage of toddlers from low-SES monolingual and language-minority immigrant families, but frequent and of long-duration in the majority of them. These results suggest contrasting parental beliefs on child-rearing. In particular, contrary to our expectation, children from language-minority immigrant families interacted with digital media and watched TV for longer than monolingual children.

What factors may account for toddlers from immigrant families watching TV and playing with a tablet for such long periods of time? First, during interviews on home language activities, some Romanian mothers reported that they regarded watching TV as a convenient source of learning the majority language, available in the home environment. Second, parental beliefs on child-rearing may also account for a longer duration of activities without an adult for toddlers from immigrant families than from monolingual families. Indeed, almost half of the immigrant families involved in this study are first-generation immigrants from non-Western traditional rural societies, that is, sociocultural contexts where adaptive models of child-rearing emphasize children’s respect for adults as the main socialization goal [46]. Within these cultural models based on “hierarchical relatedness” [47], even young children are expected to be obedient and able to stay on their own without “disturbing” parents in their chores. In addition, the parents’ knowledge of infant development, which is affected both by parents’ education and their ethnicity [2], may contribute to parenting practices. 

### 4.2. Home Language Activities in Interaction with Adults Predict Children’s Expressive Vocabulary in Low-SES Monolingual and Language-Minority Immigrant Families

In line with our expectations, the frequency and duration of language activities involving interaction with an adult in total at 24 months accounted for expressive vocabulary in total (minority + majority language) and in the majority language at 30 months, after controlling for the effect of group and total expressive vocabulary at 24 months. Neither language activities with passive exposure to language input (watching TV) nor activities involving interaction with digital media at 24 months were associated with expressive vocabulary in total or in the majority language at 30 months.

The unique contribution of measures of language activities involving interaction with an adult in total to later expressive vocabulary indicates that activities carried out in both the minority language, that is a language spoken by native speakers, and the majority language should be taken into account in promoting language acquisition in young children from language-minority immigrant families. This result is in line with work showing that the benefits of shared book-reading for preschoolers’ language and literacy skills are independent of SES [3,7]. Our results extend this work by showing that, if both languages are considered, these benefits are also independent of dual language exposure. 

Our findings highlight the importance for the promotion of expressive vocabulary acquisition of involving children from low-SES families in language activities such as shared book-reading, oral story-telling, and singing, extending previous evidence that analyzed the relation between home language activities and receptive vocabulary [3,7]. In addition, the importance of language activities involving interaction with an adult in total language used emerged for expressive vocabulary both in total language used and in the majority language. As is widely recognized, these types of home language activities increase the incidence of moments of joint attention and the richness of the linguistic input to which the children are exposed, in terms both of lexical diversity and grammatically rich constructions [11,19,27]. We speculated that these factors might in turn affect what Cummins [48] defined as ‘Common Underlying Language Proficiency’ (CUP). According to Cummins [48], every language contains surface features, but under these surface characteristics there are cognitive and linguistic competencies that are common to all languages, and it is these competencies that constitute the CUP. Conceptual knowledge of the words is common to different languages, and increasing conceptual knowledge in one or the other language may favor vocabulary acquisition in both languages. Evidence of the importance of working on cognitive and linguistic competencies that are deemed to be part of the CUP is provided by the fact that a recent intervention, combining shared book-reading and vocabulary instruction, increased vocabulary knowledge in low-SES preschool children [49].

Our results are only partially in line with recent studies that found measures of the quality of maternal speech during shared book-reading interactions, but not the frequency of reading, to account for children’s vocabulary acquisition [7,12]. Two differences, however, are that these studies considered measures of receptive vocabulary or both expressive and receptive language and that their children’s ages were different from those of our participants. 

The group as a control had an effect on expressive vocabulary in the majority language but not on total expressive vocabulary. Consistent with previous literature, therefore, the measures of bilingual children’s total language growth were equal to or greater than measures of monolingual children’s growth in their language [50]. The other control variable, children’s total expressive vocabulary at 24 months, was the most powerful predictor of expressive vocabulary at 30 months, not only in total language but also in the majority language. This finding shows that vocabulary knowledge in both languages is related and knowledge of one language supports the acquisition of the other [3]. 

Our findings concerning the role of activities with passive exposure to language input (watching TV) extend to equivalent low-SES children the null results of previous studies [3,15]. It is worth noting, however, that we did not take into account the types of programs watched by children (e.g., entertainment or educational programs). More importantly, we believe that caution is warranted on conclusions on the null effects of watching television because the children in the present study, especially children coming from language-minority families, watched TV frequently and for long periods.

Finally, our findings do not support the existence of a relationship between activities in interaction with digital media and vocabulary acquisition, and are not in line with the existing literature on groups of children at risk of language difficulties for different factors (e.g., low-SES, immigration, language delay) [25]. Unlike previous studies, however, the present study did not analyze the content and the nature of interactions with digital media [24,25].

## 5. Conclusions

The present study will help scholars disentangle the effects of low-SES and minority-majority language status in the influence of home language activities on children’s vocabulary acquisition. Our results show that children from low-SES immigrant families are as likely as children from low-SES monolingual families to be involved in a range of activities involving interaction with an adult and for a similar duration, taking together both minority and majority languages. Moreover, activities involving interaction with an adult uniquely account for expressive vocabulary acquisition in total and in the majority language in both groups.

The study has a number of limitations. First, this study compared home language activities in relatively small groups of low-SES bilingual and monolingual families. Second, groups that were different in terms of culture and ethnicity were considered as a single group of language-minority immigrant families. Therefore, future studies should not only confirm the results by considering a larger group of children from low-income families but, even more importantly, take into account the differences among cultural and ethnic groups [3], in order to gain a more comprehensive understanding of how these factors may account for differences between children from low-SES monolingual and bilingual families. Third, the study assessed frequencies and duration of home language activities, but no information on the ways in which parents interact with their children during these activities. In addition, the measure of TV-watching did not distinguish between types of TV programs watched by the children and the measure of the duration of shared book-reading did not distinguish between activities carried out in the majority language and those in the minority language. Future investigations should confirm the findings of the present study by considering more fine-grained measures of home language activities and the duration of shared book-reading in the two languages. Fourth, short versions of the CDI aimed at assessing expressive vocabulary skills in the minority language were developed for the participants of the present study by translating the short version of the Italian CDI without validating it with a norm group (this process would have taken years). Therefore, although particular care was taken to develop parallel word-lists and assessment procedures, we have to consider that these measures might not have captured the total vocabulary size in the minority language. Finally, this study focused on expressive vocabulary as an outcome measure of children’s language since previous studies mainly focused on receptive vocabulary. Future studies, however, should consider a wider range of children’s expressive and receptive language measures [7,12].

Despite the above limitations, this study has several implications. At a theoretical level, it contributes to knowledge about proximal environmental factors accounting for vocabulary acquisition in children from low-SES families. At a methodological level, it highlights the importance of considering not only total vocabulary but also total home language activities involving interaction with an adult while assessing vocabulary size and the influence of proximal environmental factors in children from low-SES language-minority families. At a practical level, it emphasizes the importance of involving children from low-SES families in a range of home language activities involving interaction with an adult frequently, for as long as possible, and as early as possible. Considering a range of language activities, rather than focusing on one type (e.g., shared book-reading), may help low-SES language-minority immigrant parents to compensate for the fact that their language activities are carried out mainly or exclusively in one language.

## Figures and Tables

**Table 1 ijerph-18-00296-t001:** Children’s and mothers’ characteristics.

	Bilinguals		Monolinguals			
	%	*M* (*SD*) Range	%	*M* (*SD*) Range	Test between Groups	*p*
Child gender (female vs. male)	57		46		χ^2^(1) = 0.72	0.46
Birth order (firstborn vs. no firstborn)	33		32		χ^2^(1) = 0.01	0.95
Singleton (yes vs. no)	100		96		χ^2^(1) = 1.94	0.16
Age of entry to nursery school (months)		14.24 (4.86) 5–23		13.50 (4.58) 7–23	*t*(68) = 0.62	0.54
Daily attendance at nursery school (hours)		7.26 (1.63) 3–10		6.63 (1.34) 3.5–8.5	*t*(68) = 1.74	0.09
Maternal age (years)		33.84 (5.30) 26–46		34.96 (4.60) 27–42	*t*(64 ^a^) = −0.84	0.41
Maternal education (more vs. fewer than 13 years/high school)	60		78		χ^2^(1 ^b^) = 2.27	0.13

Note. ^a^ Two missing data in each group. ^b^ One missing data in each group.

**Table 2 ijerph-18-00296-t002:** Weekly frequency and duration (in hours) of home language activities by the group.

	Bilinguals		Monolinguals
	Total (MinL + MajL)	MajL	Total = MajL
Variable	*M* (*SD*) Range	*M* (*SD*) Range	*M* (*SD*) Range
Frequency			
Activities in interaction with an adult ^a^	4.15 (1.85) 0.67–7	3.09 (1.85) 0–7	4.75 (1.71) 1.33–7
Shared book-reading	3.89 (2.84) 0–7	3.40 (3.02) 0–7	4.02 (2.86) 0–7
Oral story-telling	2.64 (2.74) 0–7	1.15 (2.20) 0–7	3.71 (2.92) 0–7
Singing	5.92 (2.23) 0.5–7	4.72 (3.07) 0–7	6.52 (1.64) 0.5–7
Activities with passive exposure to language input ^b^	6.24 (2.20) 0–7	6.24 (2.20) 0–7	4.67 (3.37) 0–7
Activities interacting with digital media ^c^	4.72 (3.32) 0–7	3.35 (3.53) 0–7	2.33 (3.37) 0–7
Duration			
Activities in interaction with an adult ^d^	1.00 (1.21) 0–5.16		1.10 (1.19) 0–3.50
Activities with passive exposure to language input ^b^	8.90 (6.55) 0–24.50	8.90 (6.55) 0–24.50	5.10 (4.81) 0–14.00
Activities interacting with digital media ^c^	5.49 (6.69) 0–24.50	4.62 (6.76) 0–24.50	2.77 (6.34) 0–28.00

Note. MinL = minority language; MajL = majority language. ^a^ Average value for shared book-reading, oral story-telling, and singing. ^b^ Watching TV. ^c^ Playing with smartphone and tablet. ^d^ Shared book-reading.

**Table 3 ijerph-18-00296-t003:** Mean, standard deviation, and range for expressive vocabulary size (number of words) at 24 and 30 months by the group.

	Bilinguals						Monolinguals					
	24 Months			30 Months			24 Months			30 Months		
	*M*	*SD*	Range	*M*	*SD*	Range	*M*	*SD*	Range	*M*	*SD*	Range
Total Language	30.37	27.63	0–102	69.17	4.24	6–163	37.17	24.83	3–83	61.88	25.40	3–100
Majority Language	13.52	13.93	0–52	36.50	23.36	1–97	37.17	24.83	3–83	61.88	25.40	3–100

**Table 4 ijerph-18-00296-t004:** Partial correlations (controlling for group) between home language activities and expressive vocabulary at 24 and 30 months.

	**Total Language: Frequency**			**Total Language: Duration**						
	**IA ^a^**	**PEL ^b^**	**IDM ^c^**	**IA ^d^**	**PEL ^b^**	**IDM ^c^**	**MFE**	**E_Voc Tot 24**	**E_Voc Tot 30**	**E_Voc MajL 30**
IA^a^	-	0.19	−0.06	0.62 **	0.23	0.02	0.35 **	0.01	0.21	0.25 *
PEL ^b^		-	0.08	0.14	0.58 **	0.11	0.19	0.11	0.11	0.08
IDM ^c^			-	−0.12	−0.03	0.56 **	0.19	0.05	0.05	−0.05
IA ^d^				-	0.15	−0.03	0.39 **	0.09	0.26 *	0.27 *
PEL ^b^					-	0.16	0.20	−0.17	0.10	0.03
IDM ^c^						-	0.10	−0.07	0.02	−0.05
	**Majority Language: Frequency**			**Majority Language: Duration**						
	**IA ^a^**	**PEL ^b^**	**IDM ^c^**	**IA**	**PEL ^b^**	**IDM ^c^**	**MFE**	**E_Voc Tot 24**	**E_Voc Tot 30**	**E_Voc MajL 30**
IA ^a^	-	0.15	−0.02		0.13	0.07	0.25 *	−0.07	0.02	0.14
PEL ^b^		-	0.08		0.55 **	0.09	0.19	0.11	0.11	0.08
IDM ^c^			-		−0.01	0.69 **	0.14	0.09	−0.01	−0.11
PEL ^b^					-	0.21	0.13	−0.16	0.11	−0.01
IDM ^c^						-	0.17	−0.08	0.03	−0.08
MFE							-	0.18	0.18	0.21
E_Voc Tot 24								-	0.58 **	0.60 **
E_Voc Tot 30									-	0.84 **

Note. IA = activities in Interaction with an Adult; PEL = activities with Passive Exposure to Language input; IDM = activities in Interaction with Digital Media; MFE = Maternal Formal Education; E_Voc Tot 24 = Expressive Vocabulary in Total at 24 months; E_Voc Tot 30 = Expressive Vocabulary in Total at 30 months; E_Voc MajL 30 = Expressive Vocabulary for Majority Language at 30 months. ^a^ Average value for shared book-reading, oral story-telling and singing. ^b^ Watching TV. ^c^ Playing with smartphone and tablet. ^d^ Shared book-reading. ** *p* < 0.01; * *p* < 0.05.

**Table 5 ijerph-18-00296-t005:** Hierarchical multiple regression analyses predicting total expressive vocabulary at 30 months.

	First Regression				Second Regression			
	Adjusted R^2^	Adjusted ∆R^2^	Β (SE)	*ß*	Adjusted R^2^	Adjusted ∆R^2^	Β (SE)	*ß*
Model 1	0.32 **	0.32 **			0.32 **	0.32 **		
Intercept			56.92 (11.71)				55.32 (11.82)	
Group			−12.94 (7.93)	−0.16			−12.32 (7.96)	−0.15
Expressive vocabulary in total language at 24 months			0.82 (0.14)	0.58 **			0.84 (0.14)	0.59 **
Model 2	0.36 **	0.04 *			0.36 **	0.04 *		
Intercept			11.47 (8.09)				49.93 (11.80)	
Group			4.71 (5.41)	−0.19 ^†^			−12.75 (7.76)	−0.16
Expressive vocabulary in total language at 24 months			0.82 (1.22)	0.58 **			0.81 (0.14)	0.57 **
Activities in interaction with an adult in total ^a^			0.75 (0.42)	0.21 *			6.56 (3.10)	0.21 *

Note. ^a^ Average frequency of shared book-reading, oral story-telling and singing (first regression); duration of shared book-reading (second regression). Multicollinearity test for predictors of the two regressions: 1.02 < Variance Inflation Factor < 1.04. ** *p* < 0.01; * *p* < 0.05; ^†^
*p* = 0.05.

**Table 6 ijerph-18-00296-t006:** Hierarchical multiple regression analyses predicting expressive vocabulary in the majority language at 30 months.

	Third Regression				Fourth Regression			
	Adjusted R^2^	Adjusted ∆R^2^	Β (SE)	*ß*	Adjusted R^2^	Adjusted ∆R^2^	Β (SE)	*ß*
Model 1	0.47 **	0.47 **			0.47 **	0.47 **		
Intercept			−1.57 (7.27)				−1.20 (7.39)	
Group			21.71 (4.93)	0.39 **			21.57 (4.98)	0.39 **
Expressive vocabulary in total language at 24 months			0.54 (0.88)	0.54 **			0.54 (0.09)	0.54 **
Model 2	0.52 **	0.04 **			0.50 **	0.03 *		
Intercept			−13.40 (8.29)				−4.74 (7.35)	
Group			19.71 (4.79)	0.35 **			21.29 (4.93)	0.38 **
Expressive vocabulary in total language at 24 months			0.54 (0.09)	0.54 **			0.52 (0.09)	0.52 **
Activities in interaction with an adult in total ^a^			1.12 (0.42)	0.22 **			4.31 (1.93)	0.19 *

Note. ^a^ Average frequency of shared book-reading, oral story-telling and singing (third regression); duration of shared book-reading (fourth regression). Multicollinearity test for predictors of the two regressions: 1.01 < Variance Inflation Factor < 1.04. ** *p* < 0.01; * *p* < 0.05.

## Data Availability

The data presented in this study are available on request from the corresponding author. The data are not publicly available due to privacy.

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
