# Peer review of "Home Language Activities and Expressive Vocabulary of Toddlers from Low-SES Monolingual Families and Bilingual Immigrant Families"

_ijerph, 2021, doi:10.3390/ijerph18010296_

Round 1
Reviewer 1 Report
This study investigates the impact of different home language activities on language acquisition at two different pre-school ages, comparing children from monolingual and bilingual (immigrant) low-SES families. The most original aspect of this study is the inclusion of control data of monolingual children from a comparable socio-economic background, thus enabling the authors to exclude the noise caused by socio-economic differences between the participating families.
While there are some minor shortcomings in the data collected (certain categories would better have been subdivided, allowing for a more fine-grained analysis), these limitations are acknowledged in the final section. The presentation of the data and the analysis are generally sound (with a couple of minor issues to be resolved), and both the discussion and the conclusions are insightful and make it clear that this is, indeed, an original contribution to the field.
I recommend this article for publication after minor revision, taking into account the comments below.
Specific points:
Lines 10 and 36:
“low quality”, “ language of lower quality”: The use of such evaluative terms should be avoided, as the is no such thing as “better” or “worse” language from a linguistic point of view. Instead, mores specific descriptive terms should be used, such as “less complex language”, “less varied/diverse language”, “a narrower range of syntactic structures/lexical range”, etc.
Line 74:
“interaction with digital media (watching/playing with handheld or mobile devices such as smartphone/tablet)”: I do wonder to what extent the use of a smartphone is really interactive when used by a 2-year-old. Mostly, children in that age group simply watch video clips on these handheld devices, which is really pretty much the same type of passive exposure as watching TV. If this is not the case, it would be important to describe what kind of genuinely interactive activity these 2-year-olds engage in when using a smartphone.
Lines 107/108:
Is the comparison really between low-SES parents and higher-SES mothers? If that’s what is being compared, it would be a rather skewed comparison.
Line 239:
“No significant differences were found”: No significance threshold has been mentioned. What is it?
Lines 279/280:
“... shared book-reading [...] was an index of activities carried out in both languages.”: Why? It would appear probable for book reading to be linked more closely to one of the two languages, namely the language the book is written in, than other activities like storytelling, which allow for easier switching between the two languages.
Table 2, first column (Activities in interaction with an adult):
How can the number of days (per week) on which bilingual children engage in these activities in the minority+majority language be lower for any child (0) than the number of days they engage in them in the majority language (0.67)? The child that engages in interactive activities 0.67 times a week in the majority language also engages in interactive activities in either the minority or the majority language the same number of times, doesn’t it?
Also, if there is no child that engages in singing fewer than 0.5 times a week, then how can there be any child with a combined book-reading, story-telling and singing frequency of 0 times per week.
The figures in this table should be checked for correctness.
Lines 340-342:
“Indeed, the majority of bilingual (89%) and monolingual (67%) children watched TV in the majority language almost every day, while the remaining 11% of bilinguals and 33% of monolinguals never watched TV.”
These numbers appear to exclude the possibility of watching TV in the minority language, which is, in fact widespread in the immigrant communities, due to the availability of Satellite TV in the minority languages in most immigrant homes these days.
Also, was there not a single case of a child watching TV occasionally, but not “almost every day”? It seems quite unlikely that only the two extremes, “almost every day” and “never” should apply to all 70 children.
Section 4.1., first paragraph, lines 446-457:
The discussion as to why native speakers of a minority language are more likely to interact with their children in the minority language shows a lack of understanding of immigrant psychology. The minority speakers mostly interact with their children in the minority language not so much due to a lack of proficiency in the majority language (though that may also be one contributing factor), but because the language spoken at home reflects their cultural identity, and because they feel that it is their duty to transmit the heritage language in the face of an environment that is hostile to its use. Many migrant parents speak to their children in the majority language in public, where they feel embarrassed to speak in their native language, but consciously or subconsciously avoid using the majority language at home.
Line 541, “risk group”:
Why is this term introduced here? Who is at risk of what? I suggest picking a different term.
Language and style:
Lines 10 and 27, “at risk for”: “at risk of/from”, but not “for”
Line 13, “this study analyzed”: The present tense is normally used when describing scientific studies (“this study analyzes”). Throughout the article, there is inconsistency in the use of present and past tense (e.g. line 552: “The study has a number of limitations. First, this study compared...”).
Line 57: “languages activities” > ”language activities”
Line 79: “time [they] spend in these activities” > “time [they] spend on these activities”
Line 80, “child and contingent”: What does this mean? Unclear, please rephrase.
Line 87: “Findings are mixed on the effects of watching TV on language input and acquisition.” > “Findings are mixed on the effects of watching TV on language input and acquisition.” > “Findings regarding the effects of watching TV on language input and acquisition are mixed.”
Line 89: “viewing TV” > ”watching TV”
Line 104: “that they are not distracting and did not disrupt” > “that they are not distracting and do not disrupt”
Lines 156/157, “the frequency of story-telling and reading [...] accounted for receptive vocabulary”: “correlated with”?
Line 161 and 165: “Scheele and colleagues” > “Scheele et al.”, “Takacs and colleagues” > “Takacs et al.”
Line 162: “both” > ”either”
Line 167, “computer, tablets”: Pick either the singular or the plural for both items.
Table 1: “Age of entry at nursery school” > ” Age of entry to nursery school”
Line 337: “9% of them was never involved” > ”9% of them were never involved”
Line 392: “was not correlated neither with... nor” > “was correlated neither with... nor”
Line 477: “Indeed, almost the half of” > “Indeed, almost half of”
Line 495: “talked” > “spoken”
Line 497 “early work”: Why “early”?
Line 525: “Consistently with previous literature, ...” > “Consistent with previous literature, ...”
Line 542: “more firm conclusions” > “firmer conclusions”

Reviewer 2 Report
First of all, I would like to congratulate the authors for this interesting and current work on a topic that has been little studied and is relevant in our current context.
Some aspects to be considered by the authors we have in the abstract, it would be interesting to place the type of research, instrument ..., that is, brief data on the research.
With respect to the theoretical framework, it is well constructed and updated.
The sample is correct in size, selection, and description of
characteristics of the research subjects.
The type of research is not detailed, nor is the type of
of research method used, as well as the selection of the
techniques and instruments as a consequence of all this.
Demographic questionnaire that seems to be a semi-structured interview there is no discussion of the process of constructing the interview, the validity of content.
An expressive vocabulary list is used, but is it validated?
Is this list reliable?
It must be assumed, because it is not explained, that the list of words is statistical study, it would be good to talk about the consequences which has the type of asymmetry and kurtosis, nor how it was decided
using anova, the data distribution is supposed to be normal, how do you know this?
The Bonferroni test is a multiple comparison test, it allows us to compare the means of the t levels of a factor after rejecting the null hypothesis of equality of ANOVA averages, it would be good to justify this.
The correlation that was made is not explained, was it the P-pearson, or Rho of spearman? the p of pearson is supposed to be.
With all this, my congratulations to the authors for their work, it is a good article and some very interesting results.
